# Positive In Vitro Effect of ROCK Pathway Inhibitor Y-27632 on Qualitative Characteristics of Goat Sperm Stored at Low Temperatures

**DOI:** 10.3390/ani14101441

**Published:** 2024-05-12

**Authors:** Yongjie Xu, Shixin Sun, Yu Fu, Lei Wang, Chunhuan Ren, Yinghui Ling, Zijun Zhang, Hongguo Cao

**Affiliations:** 1College of Animal Science and Technology, Anhui Agricultural University, Hefei 230036, China; yongjie@stu.ahau.edu.cn (Y.X.); 2185028577@stu.ahau.edu.cn (S.S.); fuyu@stu.ahau.edu.cn (Y.F.); 921706837@stu.ahau.edu.cn (L.W.); renchunhuan@ahau.edu.cn (C.R.); lingyinghui@ahau.edu.cn (Y.L.); zhangzijun@ahau.edu.cn (Z.Z.); 2Anhui Province Key Laboratory of Local Livestock and Poultry Genetic Resource Conservation and Bio-breeding, Anhui Agricultural University, Hefei 230036, China

**Keywords:** goat, semen, Y-27632, low-temperature preservation, metabolic mechanism

## Abstract

**Simple Summary:**

Y-27632, as a pyrimidine derivative, is commonly used for low-temperature preservation of cell cytoskeleton. However, the Y-27632 small molecule has not yet been used in research on low-temperature preservation of goat semen. This study aims to address the issue of low temperature-induced loss of sperm motility in goats by using Y-27632, and explore the regulation of Y-27632 on goat sperm metabolism. We evaluate the protective effect of Y-27632 on the quality of low temperature-preserved goat semen by detecting goat sperm motility, antioxidant capacity, mitochondrial activity, cholesterol levels, and metabolomics analysis. This study indicates that Y-27632 improves the preservation quality of goat sperm by protecting the sperm plasma membrane, enhancing sperm antioxidant capacity, and increasing D-glutamine and D-glutamate metabolism levels. The protective effect of Y-27632 on semen has also been confirmed through artificial insemination. The results of this study provide some new insights into the application of Y-27632 in low-temperature preservation of goat semen, and promote the application of Y-27632 in low-temperature preservation of semen.

**Abstract:**

Y-27632, as a cytoskeleton protector, is commonly used for low-temperature preservation of cells. Goat sperm are prone to damage to the cytoskeleton under low-temperature conditions, leading to a loss of sperm vitality. However, the Y-27632 small molecule has not yet been used in research on low-temperature preservation of goat semen. This study aims to address the issue of low temperature-induced loss of sperm motility in goats by using Y-27632, and explore the regulation of Y-27632 on goat sperm metabolism. At a low temperature of 4 °C, different concentrations of Y-27632 were added to the sperm diluent. The regulation of Y-27632 on the quality of low temperature-preserved goat semen was evaluated by detecting goat sperm motility, antioxidant capacity, mitochondrial activity, cholesterol levels, and metabolomics analysis. The results indicated that 20 µM Y-27632 significantly increased plasma membrane integrity (*p* < 0.05), and acrosome integrity (*p* < 0.05) and sperm motility (*p* < 0.05), increased levels of superoxide dismutase (SOD) and catalase (CAT) (*p* < 0.01), increased total antioxidant capacity (T-AOC) (*p* < 0.05), decreased levels of malondialdehyde (MDA) and reactive oxygen species (ROS) (*p* < 0.01), and significantly increased mitochondrial membrane potential (MMP). The levels of ATP, Ca^2+^, and TC in sperm increased (*p* < 0.01). Twenty metabolites with significant differences were identified, with six metabolic pathways having a significant impact, among which the D-glutamic acid and D-glutamine metabolic pathways had the most significant impact. The artificial insemination effect of goat semen treated with 20 μM Y-27632 was not significantly different from that of fresh semen. This study indicates that Y-27632 improves the quality of low-temperature preservation of sperm by protecting the sperm plasma membrane, enhancing sperm antioxidant capacity, regulating D-glutamine and D-glutamate metabolism, and promoting the application of low-temperature preservation of semen in artificial insemination technology.

## 1. Introduction

Y-27632, as a pyrimidine derivative, is a selective Rho-related curled coil kinase (ROCK) inhibitor. Y-27632 inhibits its kinase activity by targeting the catalytic site of ROCK-1/2 [1,2]. Research has found that Y-27632 plays an important role in cytoskeletal protection [3,4]. Y-27632 mediates RhoA activation of actin by affecting myosin light chain phosphorylation. As an important component of the cytoskeleton, actin is crucial for maintaining cytoskeleton stability [5,6]. Y-27632 can also regulate the expression level of glutathione (GSH) in cells, and the level of GSH is involved in regulating the stability of the cytoskeleton [7,8]. When the GSH content in cells decreases, the binding ability of actin is inhibited, leading to a decrease in the stability of the cytoskeleton [7]. Moreover, the abnormal glutamate metabolism and oxidative imbalance caused by GSH deficiency can also disrupt the cytoskeleton and lead to cell death [9].

Low-temperature preservation of semen plays an important role in the development of animal husbandry and rapid biological reproduction. However, during low-temperature storage, the sperm skeleton will be damaged to varying degrees, leading to a decrease in sperm motility or death [10]. In addition, during low-temperature storage, sperm antioxidant capacity weakens and ROS production increases [11]. Excessive ROS can reduce the activity of sperm skeleton proteins, causing changes in the structure or conformation, ultimately leading to sperm death [12,13]. The sperm skeleton plays an important role in maintaining sperm morphology, motility, capacitation, acrosome response, and plasma membrane protection [14]. Therefore, adding drugs or reagents to the sperm diluent can weaken the damage to the sperm skeleton and increase the in vitro preservation time of sperm. Y-27632, as a common cytoskeletal protector, has not yet been applied for low-temperature preservation of goat semen. This study investigated the protective effect of Y-27632 on low temperature-preserved sperm, and analyzed the metabolic regulation mechanism of Y-27632 on low-temperature preserved sperm. Meanwhile, the ROCK pathway activator arachidonic acid (AA) validated the protective effect of Y-27632 [15].

## 2. Material and Methods

### 2.1. Ethics Statement

Semen samples were taken from 6 healthy and fertile Anhui white goats (3 to 5 years old), and 150 female Anhui white goats (2 to 3 years old) were used for artificial insemination. All experimental goats were raised in Anhui Tianchangzhou Sheep Industry Co., Ltd. (Anhui, China) and received the same care and management, with free drinking water. The animals used in the experiment were all approved by the Animal Ethics Committee of Anhui Agricultural University (SYXK2021-09).

### 2.2. Dilution Preparation

The chemical reagents used in this study were all purchased from Sigma Aldrich (Beijing, China). Each 100 mL of semen dilution contained 20 mL of fresh egg yolk, 3.8 g of vitamin E, 1.26 g of fructose, 1.72 g of citric acid, 3.53 g of trimethylaminomethane, 0.5 g of vitamin C, 0.5 g of bovine serum albumin, 15,000 IU of penicillin sodium, and 15000 IU of streptomycin sulfate. After mixing the diluent, it was filtered and divided into ten equal parts. Y-27632 (Yuanye Biotechnology Co., Ltd., Shanghai, China) was added to the diluent at concentrations of 0, 10, 20, 30, and 50 μM, while AA (Yuanye Biotechnology Co., Ltd., Shanghai, China) was added to the other five diluents at concentrations of 0, 5, 25, 50, and 100 μM.

### 2.3. Semen Processing

Semen collection began in the autumn of 2022, using the pseudo vaginal method for semen collection twice a week, with a total of more than 100 semen samples collected. The minimum standard for sperm motility is 80%. Different concentrations of Y-27632 (0 μM, 10 μM, 20 μM, 30 μM, and 50 μM) and AA (0 μM, 5 μM, 25 μM, 50 μM, and 100 μM) were added to the sperm dilution solution. The sperm density was adjusted to 2 × 10^8^ spermatozooa/mL and stored at 4 °C for subsequent experiments.

The motility, membrane integrity, and acrosome integrity of sperm were tested, and the optimal concentrations of Y-27632 and AA were determined. On the 5th and 9th day of preservation, the antioxidant indicators of sperm in the optimal concentration AA and Y-27632 groups were measured, including superoxide dismutase (SOD) activity, catalase (CAT) activity, total antioxidant capacity (T-AOC), malondialdehyde (MDA) content, and reactive oxygen species (ROS) content; mitochondrial activity, including mitochondrial membrane potential (MMP) levels and ATP content; total cholesterol (TC) content; and Ca^2+^ content. Finally, on the 9th day, the experimental samples were analyzed by LC-MS/MS. The entire experimental process is shown in Figure 1.

### 2.4. Analysis of Sperm Motility

CASA (Computer-assisted semen analysis) was used to detect the motility of sperm (CASA, Songjingtianlun Biotechnology, Nanning, China). CASA parameters were set: temperature 37 °C, frame rate 60 Hz, particle area > 3 µm^2^, curvilinear (VCL) ≥ 25 µM/s, rectilinear (VSL) ≥ 15, and mean (VAP) ≥ 9 µM/s [16]. Then, 2 μL of semen was added to a pre-heated chamber slide at 37 °C (depth of 20 µm, Thermo Scientific, Waltham, MA, USA), and under a microscope (200×), five randomly selected fields of view were used to record the vitality of no less than 200 sperm in each sample.

### 2.5. Evaluation of Sperm Plasma Membrane Integrity Rate

In order to verify the integrity of the sperm plasma membrane, HOST (hypoosmotic swelling test) was used for detection. The low osmotic solution was composed of double-distilled water, fructose, and sodium citrate. After treatment with the low osmotic solution, the sperm were observed under a 400× microscope. The curled tail of the sperm was considered to have an intact plasma membrane [17]. The percentage of intact plasma membrane sperm was calculated, and at least three tests were conducted each time, resulting in an average value.

### 2.6. Detection of Sperm Acrosome Integrity Rate

The integrity of the acrosome was detected using the FITC-labeled peanut agglutinin (FITC-PNA) combined with the DAPI double-staining method. A suitable amount of semen was mixed with PBS solution in a 1:10 ratio, centrifuged at 1800 rpm for 3 min, and the supernatant was discarded. The sperm was fixed in 4% paraformaldehyde for 15 min, centrifuged, and the supernatant was discarded. Fluorescent isothiocyanate labeled peanut agglutinin (FITC-PNA) was added to prepare the sperm sample [18]. Then, 1 μL of DAPI fluorescent dye was added to stain the nucleus of the sperm, facilitating sperm localization [19], and incubated in a dark environment for 30 min. Sperm acrosome integrity was observed under a fluorescence microscope (400×). Green fluorescence indicates sperm acrosome, while blue fluorescence indicates sperm DNA.

### 2.7. Determination of T-AOC, SOD, CAT Activity, and MDA Content

The T-AOC of the sample was measured using the ABTS method using the T-AOC assay kit (A015-2-1, Nanjing Jiancheng Institute of Bioengineering, Nanjing, China). The collected semen was centrifuged (6000× *g*), and the supernatant was mixed with ABTS working solution for 6 min at room temperature. The optical density (OD) value at a wavelength of 405 nm was read by an enzyme-linked immunosorbent assay (ELISA) reader.

The activity of SOD in semen was measured using an SOD assay kit (BC 0175, Solarbio Biotechnology Co., Ltd., Beijing, China). The semen sample was mixed with the extraction solution, and then sonicated (power 20%, ultrasound time 3 s, interval 10 s, and total time 3 min). The supernatant was collected by centrifugation and mixed with SOD working solution. The absorbance of the sample was measured at 560 nm.

The CAT assay kit (BC 0205, Solarbio Biotechnology Co., Ltd.) was used to measure CAT activity based on the principle that H_2_O_2_ has a characteristic absorption peak at 240 nm. According to the operating instructions of the reagent kit, the sperm sample and working solution were mixed, and an ELISA reader was used to measure the initial absorbance value of the sample at 240 nm and the absorbance value after 1 min. The CAT activity obtained according to the calculation method in the manual was expressed in U/mL.

The MDA content in semen was evaluated using the MDA detection kit (BC 0025, Solarbio Biotechnology Co., Ltd.) through thiobarbituric acid reaction. The semen was mixed with the extract, sonicated, and centrifuged at 8000× *g* at 4 °C for 10 min. The supernatant was taken, and the absorbance of the sample at 532 nm and 600 nm was measured using an ELISA reader [18].

### 2.8. Determination of ROS Levels

The ROS content of semen samples was detected using an ROS assay kit (S0033S, Solarbio Biotechnology Co., Ltd.) containing DCFH-DA fluorescent probes. The semen was mixed with DCFH-DA fluorescent probes and incubated at 37 °C for 20 min. After washing with PBS, it was resuspended and the absorbance at a 488 nm excitation wavelength and 525 nm emission wavelength was measured using an ELISA reader. The higher the absorbance, the higher the level of ROS in the semen sample. 

### 2.9. Determination of Mitochondrial Membrane Potential

Sperm MMP was detected using a JC-1 fluorescent probe (C2006, Shanghai Beyotime Biology Technology Co., Ltd., Shanghai, China). Semen samples were mixed with the JC-1 fluorescent probe, incubated in the dark for 30 min, filtered through a mesh screen, and detected by flow cytometry (FACSCalibur, Franklin Lakes, NJ, USA). Normal cells containing red JC-1 aggregates were detected using the FL2 channel, and apoptotic cells containing green JC-1 monomers were detected using the FL1 channel. When detecting JC-1 monomers, the excitation wavelength was 490 nm, and the emission wavelength was 530 nm. When detecting JC1 polymers, the excitation wavelength was 525 nm, and the emission wavelength was 590 nm. At least 10,000 sperm were analyzed per sample, and the data obtained were analyzed using FlowJo software (v10.8.1).

The ATP content in semen was detected using an ATP detection kit (S0026, Shanghai Beyotime Biology Technology Co., Ltd.). The semen was mixed with cell lysate and centrifuged at 8000× *g* at 4 °C for 10 min. The supernatant was collected and mixed with the ATP detection working solution. The RLU value of the sample was measured using an ELISA reader, and the RLU value was proportional to the ATP content.

### 2.10. Determination of TC and Ca^2+^ Content in Sperm

The TC content detection kit (BC1980, Solarbio Biotechnology Co., Ltd.) was used to detect TC content in semen. In short, a sample of 100 μL and a working solution of 900 μL were thoroughly mixed and left at 37 °C for 15 min. The absorbance value at 500 nm was measured, and the results were expressed in μMol/dL.

The Ca^2+^ content in cells was rapidly and quantitatively detected by colorimetry using the Ca^2+^ detection kit (S1063S, Shanghai Beyotime Biology Technology Co., Ltd.). Semen and lysate were mixed and centrifuged at 4 °C at 10,000× *g* for 3 min. The supernatant was added to the test solution and mixed well. The sample was incubated at room temperature in the dark for 5 min. The absorbance of the sample at 575 nm was measured using an ELISA reader, and the Ca^2+^ content of the sample was calculated based on the standard curve.

### 2.11. Mass Spectrometric Analysis

#### Metabolite Extraction

Goat sperm were divided into control group and Y-27632 group, and each group of samples (*n* = 8) underwent non-targeted metabolomics analysis. Isotope-labeled extracts were added to the sperm sample, mixed, and rotated for 30 s. The sample was frozen and thawed three times in liquid nitrogen, sonicated in an ice water bath for 10 min, and incubated at −40 °C for 1 h. The sample was centrifuged at 10,000× *g* at 4 °C for 15 min [16]. The supernatant was collected and analyzed by LC-MS/MS using a UHPLC system (Vanquish, Thermo Fisher Scientific, Beijing, China) [20].

### 2.12. LC-MS/MS Analysis

Metabolites were separated using a mobile phase composed of 25 mmol/L ammonium acetate and 25 mmol/L ammonia solution. The UPLC BEH amide column (2.1 mm × 100 mm, 1.7 μM) was used in combination with a Thermo Q Exactive HFX mass spectrometer (Orbitrap MS, Thermo Fisher Scientific, Beijing, China) to detect the samples in both positive and negative ion modes [21].

### 2.13. Data Analyses Processing

ProteoWizard was used to analyze LC-MS data, including peak detection, extraction, comparison, and integration. ProteoWizard Metabolites were annotated using KEGG Pathway (http://www.kegg.jp/kegg/pathway.html, accessed on 15 January 2023) and HMDB databases, and SIMCA software (V16.0.2, Sartorius Stedim Data Analytics AB, Umea, Sweden) was used to establish PCA and OPLS-DA models. Differential metabolites were screened based on the *p*-value of the Student’s *t*-test < 0.05 and the VIP value of the OPLS-DA model > 1 [21]. 

### 2.14. Artificial Insemination

The experimental ewes were healthy female Anhui white goats aged 2–3 years old, weighing 45 ± 1.5 kg, with a total of 150 sheep. They were divided into three groups of 50 sheep in each group. The semen of Anhui white goats was divided into three parts, one of which was fresh semen, directly used for artificial insemination, and the other two parts were containing 20 μM Y-27632 and not containing 20 μM Y-27632. After being stored at 4 °C for 5 days, artificial insemination was performed. Each estrous ewe was injected with 0.5 mL of diluted semen.

### 2.15. Statistical Analysis 

The experimental data were analyzed using SPSS software (version 20.0, Chicago, IL, USA) for two-way ANOVA, presented in the form of mean ± standard deviation, and multiple comparisons were conducted using the Duncan method. *p* < 0.05 indicates significant differences, while *p* > 0.05 indicates insignificant differences. GraphPad Prism 8 was used for drawing.

## 3. Results 

### 3.1. Effects of Y-27632 and AA on the Quality of Goat Sperm

The effect of adding different concentrations of Y-27632 to semen on the quality of refrigerated sperm is shown in Figure 2. Compared with the control group, the 20 μM and 30 μM Y-27632 treatments significantly increased sperm motility on days 1–9 (*p* < 0.05). Compared with the other groups, the 20 μM Y-27632 treatment significantly increased sperm motility on day 9 (*p* < 0.05, Figure 2A). The results are displayed, demonstrating that the 10 μM, 20 μM, and 30 μM Y-27632 treatments showed higher sperm plasma membrane integrity on days 3, 6, and 9 (*p* < 0.05); the 50 μM Y-27632 treatment showed a protective effect on the integrity of sperm plasma membrane after day 6. In addition, treatment with 20 μM Y-27632 resulted in the highest sperm plasma membrane integrity and acrosome integrity on days 6 and 9 (Figure 2B–D).

The effects of AA on the sperm quality are shown in Figure 3. On the 2nd to 5th day of AA treatment, as the concentration increased, the decrease in sperm motility became more significant. Among them, the 100 μM AA treatment had the lowest sperm motility on the 3rd to 5th day, which was significantly different from the control group (*p* < 0.05). On the 5th day, it had already decreased to below 60% (Figure 3A). Compared with other groups, the 100 μM AA treatment significantly lowered the preservation quality of sperm on days 4 and 5 (*p* < 0.05, Figure 3B,C).

Based on the above findings, 20 μM Y-27632 showed the best effect in improving sperm quality and was selected as the experimental group for subsequent experiments, while 100 μM AA was the optimal concentration to verify the protective effect of ROCK inhibitor Y-27632 on sperm.

### 3.2. Analysis of Y-27632 and AA on Antioxidant Ability of Goat Sperm

The results showed that 20 μM Y-27632 significantly increased the SOD content, CAT content (*p* < 0.01, Figure 4A,B), and T-AOC level (*p* < 0.05, Figure 4C) of sperm (day 9). Meanwhile, the levels of MDA and ROS were reduced (*p* < 0.01, Figure 4D,E). On the contrary, 100 μM AA significantly reduced the SOD and CAT content of sperm (*p* < 0.01, Figure 4F,G), while T-AOC levels showed no significant change (*p* > 0.05, Figure 4H), and while MDA and ROS levels significantly increased (*p* < 0.01, Figure 4I,J) (day 5). In summary, 20 μM Y-27632 can enhance the antioxidant capacity of goat sperm.

### 3.3. Effects of Y-27632 and AA on Mitochondrial Energy Metabolism in Goat Sperm

MMP directly determines the survival status of cells, with mitochondria being a key site for ATP synthesis. Therefore, detecting MMP levels and ATP content can reflect the preservation effect of sperm. The high potential of the 20 μM Y-27632 group was 72.5%, higher than that of the control group (Figure 5A,B), and the ATP content significantly increased (*p* < 0.01, Figure 5C) (day 9). However, the 100 μM AA group was 14.6%, lower than the control group (Figure 6A,B), and the ATP content significantly decreased (*p* < 0.01, Figure 6C) (day 5). The data shows that 20 μM Y-27632 has a good protective effect on sperm mitochondria.

### 3.4. Effects of Y-27632 (Day 9) and AA (Day 5) on Ca^2+^ and TC in Goat Sperm

Ca^2+^ is a core regulatory factor for sperm capacitation and acrosome response, and cholesterol plays an important role in stabilizing the sperm plasma membrane. The results indicate 20 μM Y-27632 significantly increased sperm Ca^2+^ and TC content (*p* < 0.01, Figure 7A,B) (day 9). On the contrary, 100 μM AA significantly reduced sperm Ca^2+^ and TC content (*p* < 0.01, Figure 7C,D) (day 5). It can be inferred that 20 μM Y-27632 can protect sperm by regulating the levels of Ca^2+^ and cholesterol.

### 3.5. Metabolomics Analysis of Y-27632 Goat Sperm

By combining the multiple changes of metabolites with *p*-values, differential metabolites were screened and presented in the form of a volcano plot (Figure 8A). The data can be found in Appendix A, and 20 representative differential metabolites were further screened. In comparison with the control group, 10 differential metabolites such as Decanoylcarnitine, Isocaproic acid, L-Nicotine were significantly downregulated in the Y-27632 group (*p* < 0.05); 10 differential metabolites, including 7,8-Dehydro-beta-micropteroxantin and SM (d18:1/18:1 (9Z)), were significantly upregulated in Y-27632 (*p* < 0.05) (Table 1, Figure 8B and Appendix A). By analyzing the correlation between differential metabolites, we aim to elucidate the regulatory roles of metabolites in various biological processes. 2’,4’-Dihydroxycetophenone and 2-Methylbenzoic acid (r = 0.96), L-Hexanoylcarnitine and L-Octanoylcarnitine (r = 0.99), and (25S)-26-Hydroxy-24-methylenecycloartan-3-one and 2-Acetylpyrazine (r = 0.92) were positively correlated. 2’,4’-Dihydroxyacetophenone and LysoPE (20:4 (8Z, 11Z, 14Z, 17Z)/0:0) (r = −0.70), Decanoylcarnitine and Byssochlamic acid (r = −0.71), and 2,3,4,5-Tetrahydropiperidine-2-carboxylate and 9,10-Epoxyoctadecanoic acid (r = −0.88) were negatively correlated (Figure 8C). In addition, the metabolite correlation analysis between the sheep Y-27632 group and the goat Y-27632 group showed that PC(o-18:1(9Z)/20:4(8Z, 11Z, 14Z, 17Z)) was positively correlated with 2-O-(4,7,10,13,16,19-Docosahexaenoyl)-1-O-hexadecylglycero-3-phosphocholine (r = 1), and (S)-beta-Aminoisobutyric acid was positively correlated with Choline glycerophosphate (r = 0.96). However, Vaccenic acid was negatively correlated with PE(20:0/18:4(6Z, 9Z, 12Z, 15Z)) (r = −0.94) and Fructose-1P (r = −0.96) (Appendix A).

KEGG metabolic pathway analysis was conducted on differential metabolites, and ultimately enriched to 108 metabolic pathways (Appendix A). In the upregulated metabolites of sperm in the Y-27632 group, L-Glutamic acid, Isocitric acid, Pyruvic acid, and L-Glutamine were mainly annotated on several metabolic pathways, including the biosynthesis of amino acids, taurine and hypotaurine metabolism, carbon metabolism, metabolic pathways, and central carbon metabolism in cancer (Figure 9A and Table 2).

By conducting enrichment analysis and topological analysis on the pathways of differential metabolites, we have identified the key pathways with the highest correlation with differential metabolites. The results showed that differential metabolites were involved in thirty-seven pathways (Appendix A), among which six metabolic pathways were mainly affected, namely D-glutamine and D-glutamate metabolism, nitrogen metabolism, alanine, aspartate and glutamate metabolism, beta-alanine metabolism, glycerolipid metabolism, and taurine and hypotaurine metabolism, with the D-glutamine and D-glutamate metabolism pathways having the greatest impact and the most significant enrichment level (*p* < 0.05, Figure 9B and Table 3).

## 4. Fertility Analysis

The fertility results are shown in Table 4. In terms of fertility, there was no significant difference (*p* > 0.05) between semen containing 20 μM Y-27632 stored at low temperature for five days and fresh semen. However, compared with the control group without 20 μM Y-27632, the fertility rate of the control group was significantly reduced (*p* < 0.05).

## 5. Discussion

Sperm motility, plasma membrane integrity, and acrosome integrity are important indicators for evaluating sperm quality [22]. The plasma membrane and acrosome are important structures for sperm capacitation. When sperm are in a low-temperature environment, the decrease in temperature can lead to cold shock or the internal formation of ice crystals and denaturation of sperm proteins [23], which in turn makes the plasma membrane more fragile and directly affects the conception rate of sperm. The acrosome contains various hydrolytic enzymes [24]. During preservation, the acrosome is damaged, causing the release of substances in the acrosome and preventing acrosome reactions, resulting in the loss of fertilization ability of sperm. In our experiment, the sperm motility, plasma membrane integrity rate, and acrosome integrity rate of the 20 μM Y-27632 treatment on the 9th day were 68.69%, 71%, and 76%, respectively, significantly higher than those of the other groups, indicating that Y-27632 can improve the quality of sperm preservation. Y-27632 is a selective ROCK1 inhibitor that has been shown to be co-expressed in the head and tail of mammalian sperm [25]. To investigate whether Y-27632 has a protective effect on sperm by affecting the ROCK pathway, the ROCK pathway activator AA was selected for validation. The results showed that AA reduced the motility, membrane integrity, and acrosome integrity of goat sperm, and Y-27632 protected sperm by inhibiting the ROCK pathway.

ROS balance is an important factor in maintaining sperm structure and function. On the one hand, an appropriate amount of ROS is involved in sperm capacitation, acrosome reaction, and fertilization, and on the other hand, excessive ROS can lead to lipid peroxidation and DNA damage [26]. SOD and CAT are the main antioxidant enzymes in sperm, which can maintain sperm oxidative balance [27]. SOD can effectively eliminate free radicals and transform the superoxide anion O_2_^−^ into the less toxic H_2_O_2_, thereby weakening the damage caused by oxidative stress on sperm [28]. CAT can convert H_2_O_2_ into H_2_O [29], reducing the toxic effect of H_2_O_2_ on sperm. MDA is the main detection indicator of lipid peroxidation products, which mainly damages the DNA of sperm [30]. The higher the MDA content in sperm, the more severe the lipid peroxidation of sperm and the higher the degree of sperm damage [31]. The increase in oxidative stress levels is the most important cause of sperm DNA damage [26]. According to reports, oxidative stress ultimately destroys sperm DNA by incubating sperm under high oxygen pressure, thereby reducing sperm motility [32]. Therefore, the detection of the antioxidant system, ROS, and MDA content of sperm can indirectly reflect the degree of damage to sperm cells. As expected, Y-27632 reduced the content of ROS and MDA in semen and increased the activity of antioxidant enzymes, indicating that Y-27632 protects sperm from oxidative damage by enhancing antioxidant capacity.

The mitochondria of sperm mainly rely on various metabolic pathways, such as the tricarboxylic acid cycle and oxidative phosphorylation [33], to provide the necessary ATP for sperm motility. In mitochondria, normal MMP is an important condition for maintaining ATP synthesis. When sperm mitochondria are damaged, their activity decreases and the energy produced is insufficient, making it impossible for sperm to move and fertilize normally. Research has shown that Y-27632 can compete with ATP for its binding to kinases [34]. In this study, the addition of Y-27632 increased the ATP content and MMP levels in sperm. We speculate that Y-27632 regulates mitochondrial activity by inhibiting ROCK signaling, thereby reducing ATP consumption. Ca^2+^ has been found to be a core regulatory factor for sperm, capacitation, and even acrosome response [35]. The increase in Ca^2+^ in cells can enable sperm to activate and produce acrosome reactions [36]. In addition, the increase in intracellular Ca^2+^ will overactivate some pathways in sperm, causing them to become more active, consume energy too quickly, and accelerate sperm apoptosis [37]. When sperm are stored in vitro, low temperature causes the rupture of the sperm plasma membrane, which can cause extracellular Ca^2+^ to enter the sperm interior, leading to an increase in intracellular Ca^2+^ concentration and ultimately sperm capacitation [38]. Therefore, during the preservation process, inhibiting the influx of Ca^2+^ and preventing premature sperm capacitation can prolong the semen preservation time [39]. The results of this experiment showed that the Ca^2+^ concentration of 20 μM Y-27632 was significantly increased compared to the control group. We speculate that Y-27632 can reduce the damage of Ca^2+^ to sperm.

The membrane lipids of the sperm plasma membrane have an important protective effect on sperm in vitro preservation, mainly including phospholipids, sterols, and glycolipids [40]. The main component of sterols is cholesterol, which plays a stabilizing role in membranes [41]. When the temperature is too high, it can stabilize the lipid bilayer of the cell, inhibit the excessive flow of lipid molecules in the membrane, and maintain the stability of the cell membrane. When the temperature is too low, it can also improve the fluidity of the membrane [42]. Adding cholesterol has also been proven to maintain fluid state and reduce membrane damage at lower temperatures [43]. Therefore, the importance of cholesterol for membranes is evident. In this experiment, the total cholesterol content of sperm with 20 μM Y-27632 added to the goat semen dilution was significantly higher than the control group, indicating that Y-27632 can protect sperm by increasing cholesterol content. This is consistent with the research results of Batissaco et al. [44] on adding cholesterol in semen.

Sphingolipid (SM) is a very stable phospholipid [45], and phosphatidylethanolamine (PE) is the most abundant phospholipid in the cell membrane [46]. Both can maintain the integrity of the sperm plasma membrane [47]. Lucio [48] found that the difference in lipid composition of the plasma membrane directly affects the motility of sperm. PE is an important lipid marker for highly motile sperm, and the difference in plasma membrane composition is a key factor determining sperm motility, sperm function, and susceptibility to oxidative stress. Similarly, our study found that LysoPE (20:4 (8Z, 11Z, 14Z, 17Z)/0:0) and SM (d18:0/18:1 (9Z)) were higher in goat semen with the addition of Y-27632. The results showed that Y-27632 can alleviate damage to the sperm plasma membrane by increasing the content of phospholipids in the membrane. Through metabolomics analysis, the pathways with the most significant differences are the D-glutamine and D-glutamate metabolism pathways, involving different metabolites such as glutamate, D-glutamine, and glutamine. D-glutamine and D-glutamate metabolism belong to the metabolism of amino acids, and in the process of glycolysis metabolism, glutamine and glutamate can be converted into each other [49]. Glutamine plays a crucial role in cell growth and metabolism, and glutamate participates in the synthesis of glutathione (GSH) [50]. GSH can reduce oxidative stress damage to the body [51].

The role of Y-27632 in low-temperature preservation of semen has also been confirmed through artificial insemination in goats. The goat semen treated with 20 μM Y-27632 is stored at 4 °C for 5 days, and the fertility of artificial insemination is no different from that of fresh semen. This will promote the widespread application of low-temperature preserved semen in artificial insemination technology and promote the rapid development of animal husbandry.

## 6. Conclusions

During the low-temperature preservation process of goat semen, the ROCK pathway inhibitor Y-27632 increased the content of sperm ATP, SOD, CAT, TC, MMP, glutamine, and glutamate, decreased the content of ROS and MDA, and prolonged the preservation time and quality of semen (Figure 10).

## Figures and Tables

**Figure 1 animals-14-01441-f001:**
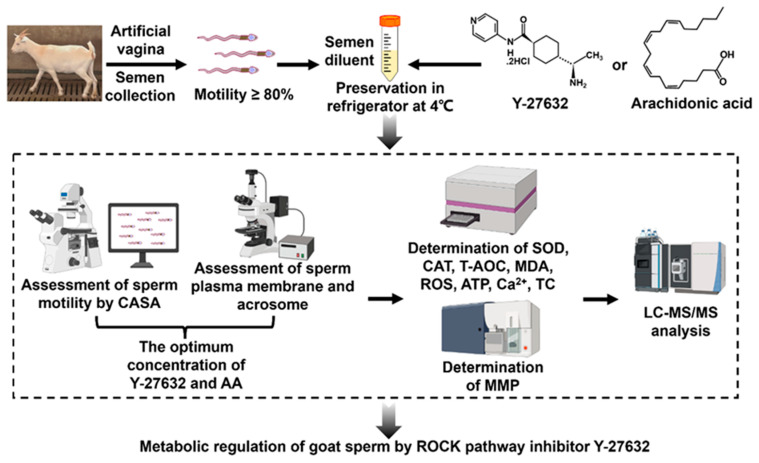
Experimental design and workflow.

**Figure 2 animals-14-01441-f002:**
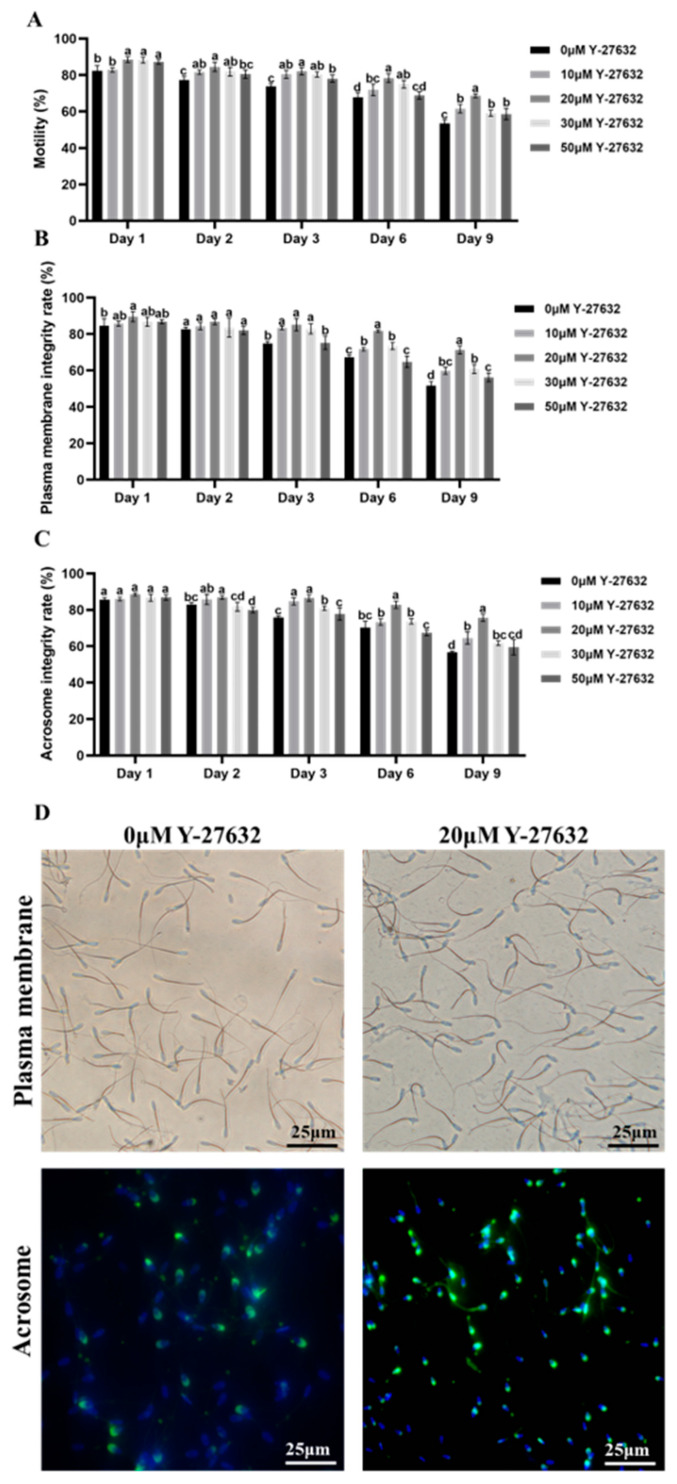
Effects of Y-27632 on quality. (**A**) Regulation of sperm motility by Y-27632. (**B**) Regulation of sperm plasma membrane integrity rate by Y-27632. (**C**) Regulation of sperm acrosome integrity rate by Y-27632 during preservation at 4 °C. Different letters on the histogram indicate significant differences between groups (*p* < 0.05). (**D**) Regulation of integrity of sperm plasma membrane and acrosome by Y-27632 (day 9).

**Figure 3 animals-14-01441-f003:**
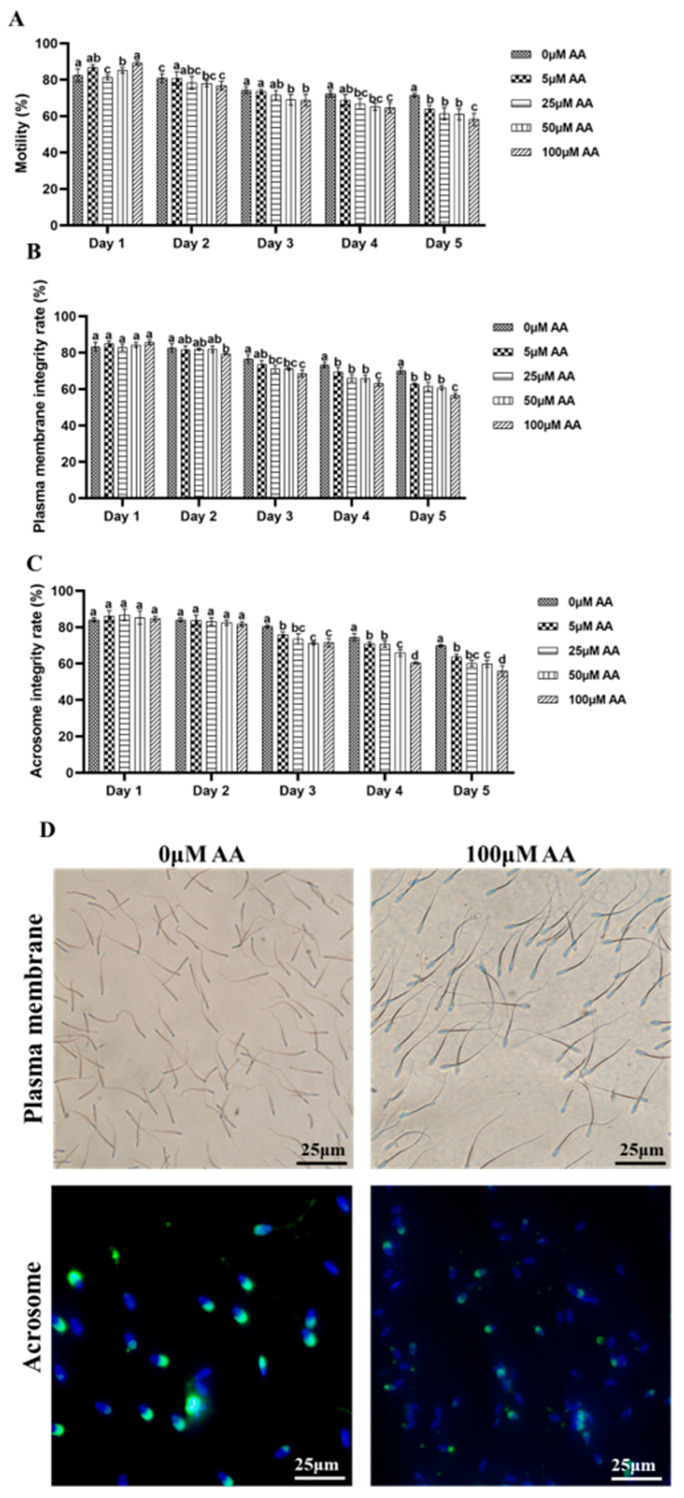
Effects of AA on sperm qlity. (**A**) Regulation of sperm motility by AA. (**B**) Regulation of sperm plasma membrane integrity rate by AA. (**C**) Regulation of sperm acrosome integrity rate by AA during preservation at 4 °C. Different letters on the histogram indicate significant differences between groups (*p* < 0.05). (**D**) Regulation of integrity of sperm plasma membrane and acrosome by AA (day 9).

**Figure 4 animals-14-01441-f004:**
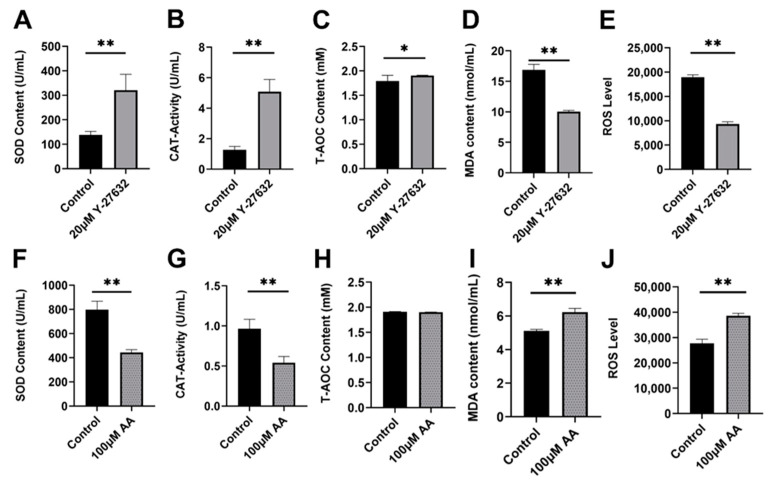
The effects of Y-27632 (day 9) and AA (day 5) on the antioxidant capacity of sperm. (**A**) Regulation of SOD content by 20 μM Y-27632. (**B**) Regulation of CAT activity by 20 μM Y-27632. (**C**) The effect of regulation of T-AOC level by 20 μM Y-27632. (**D**) Regulation of MDA content by 20 μM Y-27632. (**E**) Regulation of ROS level by 20 μM Y-27632. (**F**) Regulation of SOD content by 100 μM AA. (**G**) Regulation of CAT activity by 100 μM AA. (**H**) Regulation of T-AOC level by 100 μM AA. (**I**) Regulation of MDA content by 20 μM Y-27632. (**J**) Regulation of ROS level by 20 μM Y-27632. * indicates *p* < 0.05, ** indicates *p* < 0.01.

**Figure 5 animals-14-01441-f005:**
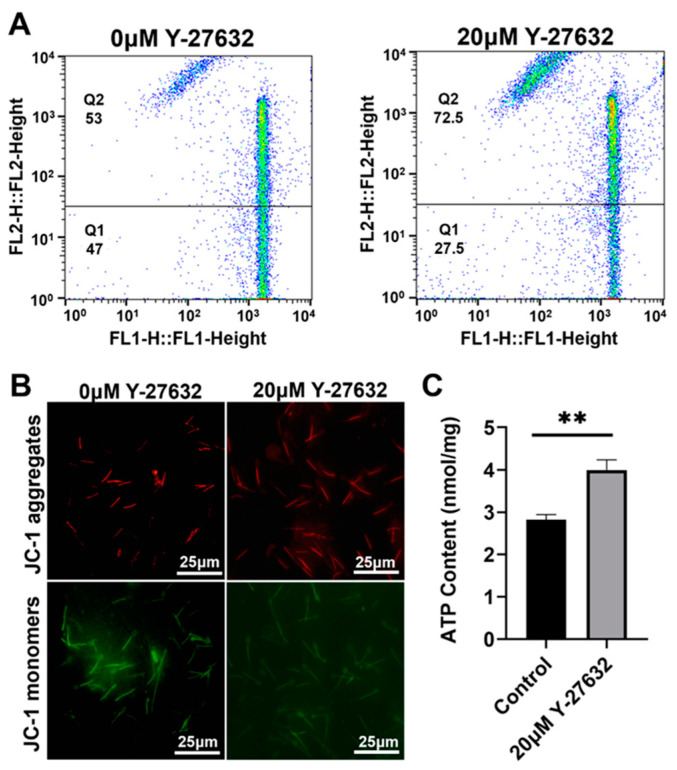
The effects of Y-27632 (day 9) on the MMP and ATP levels of sperm. (**A**) Flow cytometry detection of sperm MMP by Y-27632 treatment. (**B**) JC-1 fluorescence staining diagram. (**C**) The effect of Y-27632 on the ATP content of sperm. Sperm mitochondrial JC-1 fluorescence staining, red represents high membrane potential staining, green fluorescence represents low membrane potential staining. ** indicates *p* < 0.01.

**Figure 6 animals-14-01441-f006:**
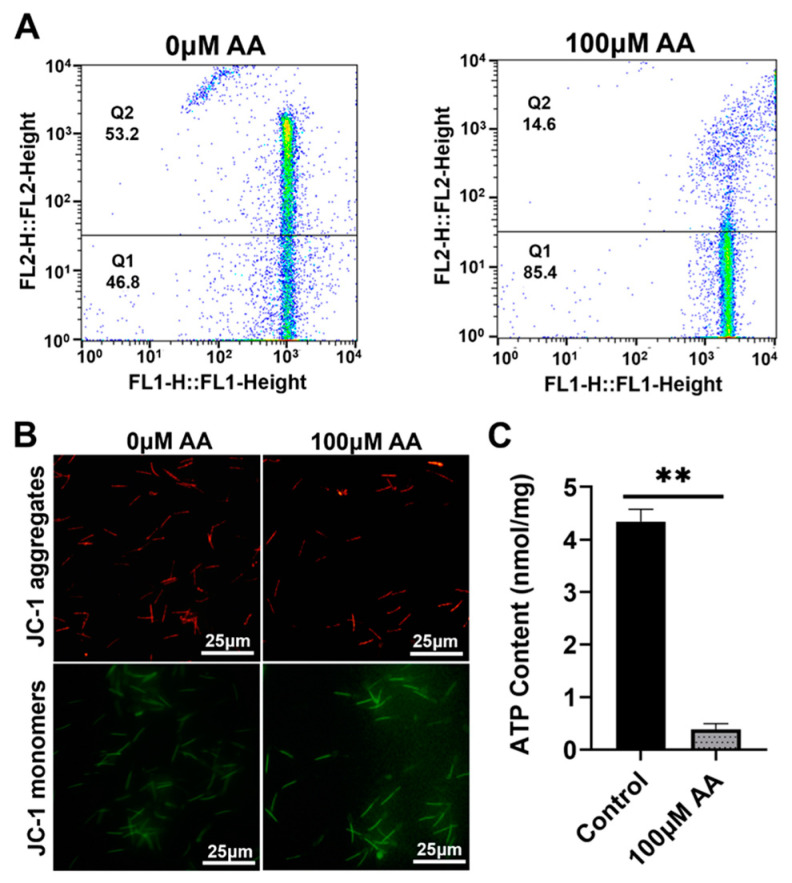
The effects of AA (day 5) on the MMP and ATP levels of sperm. (**A**) Flow cytometry detection of sperm MMP by AA treatment. (**B**) JC-1 fluorescence staining diagram. (**C**) The effect of AA on the ATP content of sperm. Sperm mitochondrial JC-1 fluorescence staining, red represents high membrane potential staining, green fluorescence represents low membrane potential staining. ** indicates *p* < 0.01.

**Figure 7 animals-14-01441-f007:**
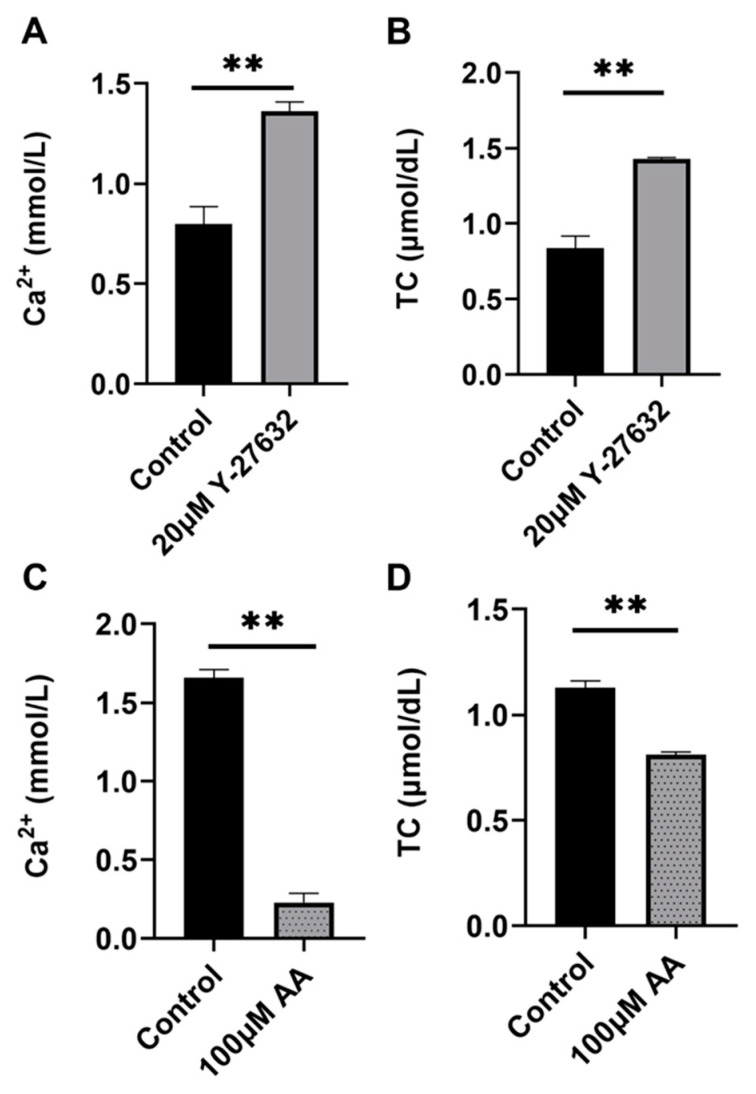
The effects of Y-27632(day 9) and AA(day 5) on the levels of Ca^2+^ and TC of sperm. (**A**) Regulation of Ca^2+^ content by 20 μM Y-27632. (**B**) Regulation of TC activity by 20 μM Y-27632. (**C**) Regulation of Ca^2+^ content by 100 μM AA. (**D**) Regulation of TC activity by 100 μM AA. ** indicates *p* < 0.01.

**Figure 8 animals-14-01441-f008:**
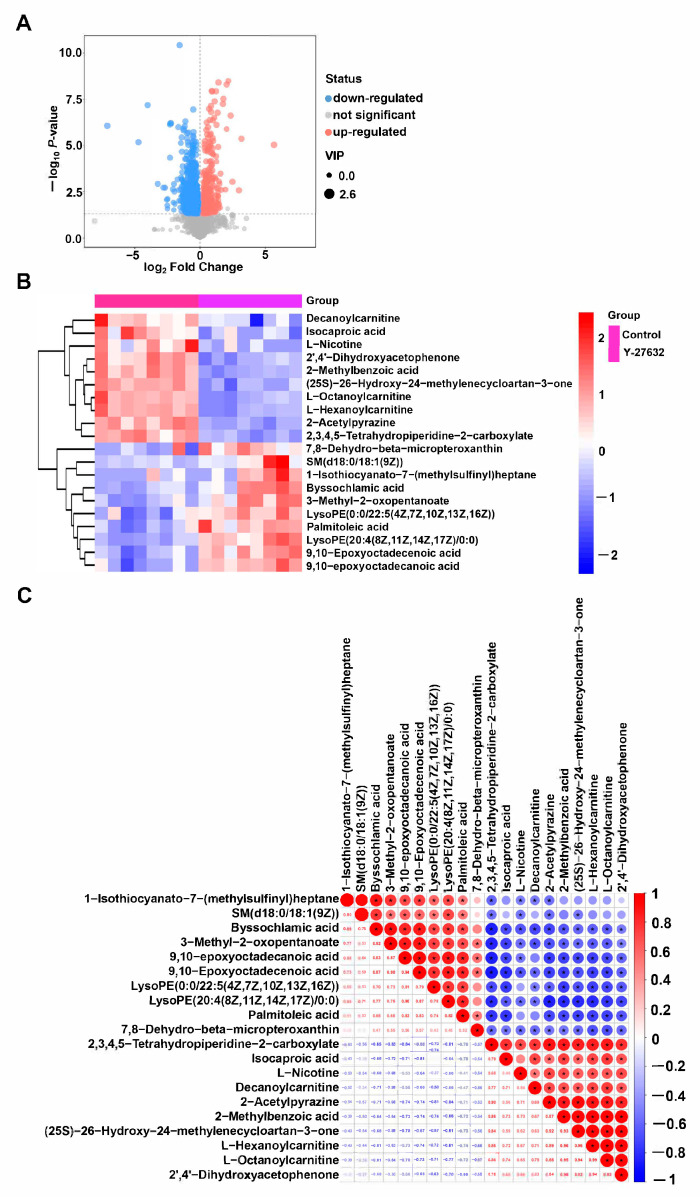
Regulation and differential metabolism of Y-27632 on goat sperm. (**A**) Volcanic map for screening different metabolites. (**B**) Differential metabolite cluster heat map. (**C**) Correlation analysis of differential metabolites. * means significance.

**Figure 9 animals-14-01441-f009:**
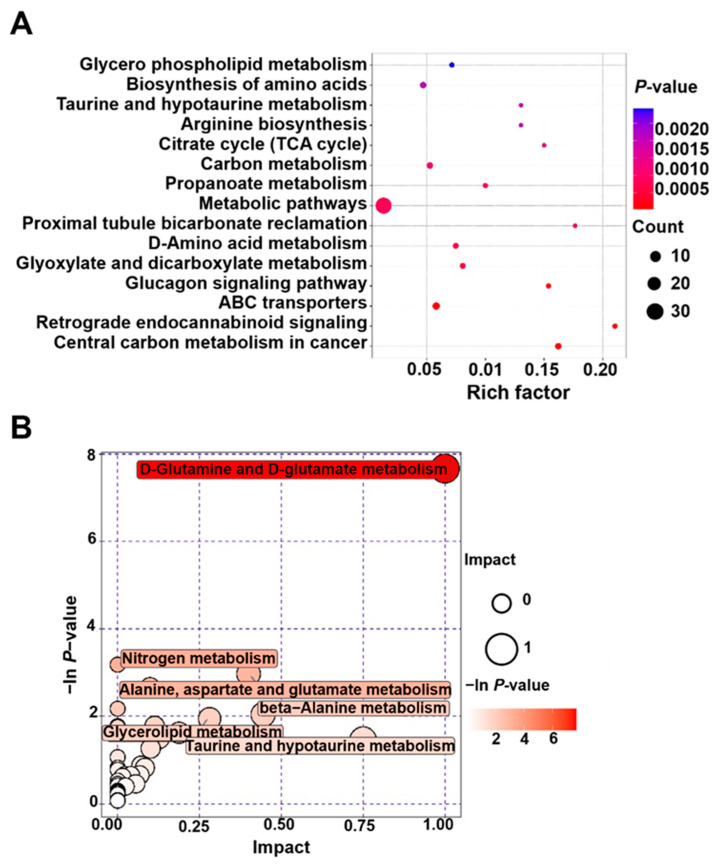
Analysis of the regulatory pathway of Y-27632 on differential metabolites in goat sperm. (**A**) Analysis of regulatory pathways of differential metabolites. (**B**) KEGG enrichment analysis of differential metabolites. Pathway analysis between control and 20 μM Y-27632 groups.

**Figure 10 animals-14-01441-f010:**
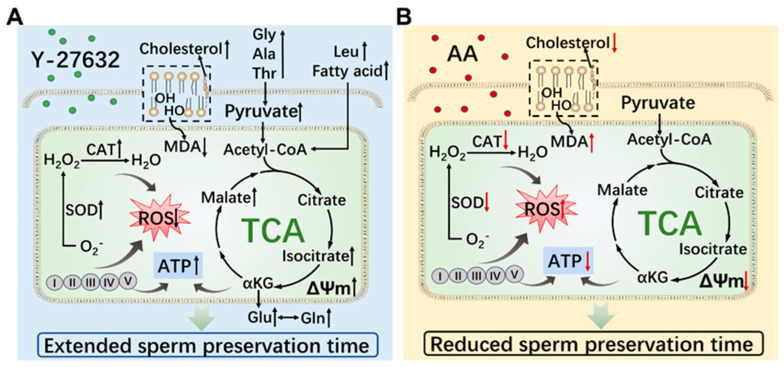
The regulatory mechanism of Y-27632 and AA on goat sperm metabolism. (**A**) The regulatory mechanism of ROCK pathway inhibitor Y-27632 on sperm metabolism in goats. (**B**) The regulatory mechanism of ROCK pathway activator AA on sperm metabolism in goats. Gly, glycine; Ala, alanine; Thr, threonine; Leu, leucine; Glu, glutamate; Gln, glutamine; MDA, malonaldehyde; CAT, catalase; SOD, superoxide dismutase; ROS, reactive oxygen species; αKG, α-Ketoglutaric acid; ΔΨm, mitochondrial membrane potential; I, II, III, IV, and V represents the respiratory chain.

**Table 1 animals-14-01441-t001:** Analysis of the metabolites of goat sperm in Y-27632 group.

Name	VIP	*p*-Value	FC
Isocaproic acid	2.10472657	0.001404849	1.640539
2’,4’-Dihydroxyacetophenone	2.360230926	0.000002668	1.706914
Decanoylcarnitine	1.626249997	0.002329103	1.707014
L-Octanoylcarnitine	2.462864583	0.000000073	1.789571
L-Hexanoylcarnitine	2.458776918	0.000000011	1.876355
(25S)-26-Hydroxy-24-methylenecycloartan-3-one	2.285585719	0.000000041	2.173294
2-Acetylpyrazine	2.44681402	0.000000012	2.336872
2-Methylbenzoic acid	2.544358081	0.000000769	2.397188
L-Nicotine	1.950733563	0.002509318	3.111003
2,3,4,5-Tetrahydropiperidine-2-carboxylate	2.464528754	0.000000057	3.380712
1-Isothiocyanato-7-(methylsulfinyl)heptane	1.198204213	0.030862102	0.248183
LysoPE(20:4(8Z, 11Z, 14Z, 17Z)/0:0)	2.326823113	0.000003663	0.376789
SM(d18:0/18:1(9Z))	1.79944416	0.036462866	0.462591
LysoPE(0:0/22:5(4Z, 7Z, 10Z, 13Z, 16Z))	1.862054009	0.000668867	0.499072
9,10-Epoxyoctadecenoic acid	2.165526982	0.000034265	0.508835
Byssochlamic acid	2.118297325	0.001568542	0.548432
Palmitoleic acid	2.132674871	0.000147640	0.548844
9,10-epoxyoctadecanoic acid	2.159295557	0.000027289	0.597584
7,8-Dehydro-beta-micropteroxanthin	1.642921181	0.010008768	0.614358
3-Methyl-2-oxopentanoate	2.101091738	0.000441510	0.650236

VIP, variable importance in projection; FC, fold change.

**Table 2 animals-14-01441-t002:** Enrichment analysis of differential metabolites in goat sperm from Y-27632 group.

Name	*p*-Value	Number
Glycerophospholipid metabolism	0.002131231	4
Biosynthesis of amino acids	0.001475389	6
Taurine and hypotaurine metabolism	0.001407924	3
Arginine biosynthesis	0.001407924	3
Citrate cycle (TCA cycle)	0.000925305	3
Carbon metabolism	0.000804546	6
Propanoate metabolism	0.000595049	4
Metabolic pathways	0.000571736	39
Proximal tubule bicarbonate reclamation	0.000563529	3
D-Amino acid metabolism	0.000466421	5
Glyoxylate and dicarboxylate metabolism	0.000324201	5
Glucagon signaling pathway	0.000107687	4
ABC transporters	0.000049142	8
Retrograde endocannabinoid signaling	0.000029368	4
Central carbon metabolism in cancer	0.000001238	6

*p*-value: the *p*-value obtained from the *t*-test of the substance in this group comparison.

**Table 3 animals-14-01441-t003:** Analysis of the metabolic pathway of goat sperm in Y-27632 group and control group.

Pathway	Total	Hits	Raw *p*	Holm Adjust	Impact
D-Glutamine and D-glutamate metabolism	5	3	0.000468	0.037905	1
Nitrogen metabolism	9	2	0.041713	1	0
Alanine, aspartate, and glutamate metabolism	23	3	0.05148	1	0.4
beta-Alanine metabolism	17	2	0.13046	1	0.44444
Glycerolipid metabolism	18	2	0.14339	1	0.28098
Taurine and hypotaurine metabolism	7	1	0.23455	1	0.75

**Table 4 animals-14-01441-t004:** Fertility parameters of semen.

Semen Type	Fertilized Ewes (*n*)	Fertility (%)
Fresh semen	50	43.2 ± 0.26 ^c^
Semen with 20 μM Y-27632	50	42.5 ± 0.15 ^bc^
Semen without 20 μM Y-27632	50	31.2 ± 0.23 ^a^

Different letters represent significant differences (*p* < 0.05).

## Data Availability

The authors confirm that the data supporting the study findings are available in the article and Appendix A.

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
