# Peer review of "Positive In Vitro Effect of ROCK Pathway Inhibitor Y-27632 on Qualitative Characteristics of Goat Sperm Stored at Low Temperatures"

_animals, 2024, doi:10.3390/ani14101441_

Round 1

Reviewer 1 Report

Comments and Suggestions for Authors

The manuscript entitled “Metabolic regulation of ROCK pathway inhibitor Y-27632 on goat sperm” had the hypothesis that Y-27632 plays an important role in sperm quality, because of its beneficial properties to the cytoskeleton. The authors executed several in-vitro analyses of sperm quality, including metabolic pathways, obtaining interesting and good results. The paper is well written, with clear material & methods, and appropriate experimental design.

There are suggestions included in the attached file.

However, I would like to highlight that the authors said that they did several sperm assessments on days 5th and 9th of sperm storage (Line 108); whereas, in the results section, it is not clear from what chilling days are the data. I suggest inserting data from both days.

Author Response

Dear Reviewers:

       We are truly grateful to you and the reviewers for the critical comments and thoughtful suggestions on our manuscript. They are really helpful and based on these comments and suggestions, we have revised the manuscript carefully. In the following pages are our point-by-point responses to the reviewers’ comments/suggestions. Please feel free to contact us if there is any question and we are very willing to improve our manuscript until all the reviewers are satisfied.

Best regards,

Hongguo Cao

Address: College of Animal Science and Technology, Anhui Agricultural University, Hefei 230036, P.R. China

E-mail: caohongguo1@ahau.edu.cn

Reviewer:1

C: However, I would like to highlight that the authors said that they did several sperm assessments on days 5th and 9th of sperm storage (Line 108); whereas, in the results section, it is not clear from what chilling days are the data. I suggest inserting data from both days.

A: Thank you for pointing out the problem. According to your comments and suggestions, we have added the number of days for sperm evaluation.

C:Insert reference.

A: Thank you for pointing out the problem. According to your comments and suggestions, we have added the corresponding references. Line74:references11

C: Avoid to start sentences with number.

A: Thank you for pointing out the problem. According to your comments and suggestions, we have made the modifications. Line96

C: The ejaculates were not pooled? Each buck ejaculate represented one experimental unit?

A: Thank you for pointing out the problem. The semen we collected was mixed for subsequent experiments. Each buck ejaculation does not represent an experimental unit.

C: G force?

A: Thank you for pointing out the problem. According to your comments and suggestions, we have added the corresponding speed. Line137

C: Don't start a sentence with numbers.

A: Thank you for pointing out the problem. According to your comments and suggestions, we have made the modifications. Line151

C: Again. Please, check it throughout the whole manuscript.

A: Thank you for pointing out the problem. According to your comments and suggestions, we have conducted verification and modification.

C: Provide information about the flow cytometer machine.

A: Thank you for pointing out the problem. According to your comments and suggestions, we have added the model of the flow cytometer. Line175

C: What was considered the experimental unit? If each ejaculate from each buck was an experimental unit, I suggest you to test if there were differences among each buck.

A: Thank you for pointing out the problem. The experimental unit refers to an independent carrier that can accept different experimental treatments.

C: The motility obtained after 9 days of storage was really high, when compared with other reports, even for the control group: Do you have an explanation for that?

A: Thank you for pointing out the problem. This experiment investigated the protective effect of Y-27632 on low-temperature preserved sperm. On the 9th day, the sperm motility of the Y-27632 group remained high and significantly higher than that of the control group, indicating that Y-27632 can indeed provide a good protective effect on low-temperature preserved sperm.

C: Correct the bar size of the pictures. It is clear that the images were taken in different magnifications. Furthermore, insert photomicrographies with higher magnification for plasma membrane, to make easier for the reader identifying sperm presenting coiled tails.

A: Thank you for pointing out the problem. According to your comments and suggestions, we have modified the image.Figure2

C: Acrosome photomicrographies have higher magnification than plasma membrane. Insert the appropriate bar size.

A: Thank you for pointing out the problem. According to your comments and suggestions, we have modified the image.Figure3

C: You said (L 108) you measured antioxidant characteristics at 5th and 9th days. In this section, you only presented one result for each parameter and treatment. Please, correct it.

A: Thank you for pointing out the problem. According to your comments and suggestions, we have added the corresponding number of experimental days. Line265,269

C: Day 5 or 9? Title; Detection of sperm MMP and ATP leves of chilled sperm treated with Y-27632.

A: Thank you for pointing out the problem. According to your comments and suggestions, we have added the corresponding number of experimental days. The Y-27632 group used semen from the 9th day of low-temperature storage, while the AA group used semen from the 5th day of low-temperature storage. Line273

C:Day 5 or 9?

A: Thank you for pointing out the problem. According to your comments and suggestions, we have added the corresponding number of experimental days. The Y-27632 group used semen from the 9th day of low-temperature storage, while the AA group used semen from the 5th day of low-temperature storage. Line284-286

C: Day 5 or 9? Title; Detection of sperm MMP and ATP leves of chilled sperm treated with Y-27632.

A: Thank you for pointing out the problem. According to your comments and suggestions, we have added the corresponding number of experimental days. The Y-27632 group used semen stored at low temperature for the 9th day. Line289

C: Correct the title, as suggested before. Day 5 or 9?

A: Thank you for pointing out the problem. According to your comments and suggestions, we have added the corresponding number of experimental days. The Y-27632 group used semen stored at low temperature for the 9th day. Line309

C:Effects of Y-27632 and AA on Ca2+ and TC in goat sperm .Day 5 or 9?

A: Thank you for pointing out the problem. According to your comments and suggestions, we have added the corresponding number of experimental days. The Y-27632 group used semen from the 9th day of low-temperature storage, while the AA group used semen from the 5th day of low-temperature storage. Line317

C:Correct the format of the table legend.

A: Thank you for pointing out the problem. According to your comments and suggestions, we have made the modifications.

C:The work cited was conducted with spermatogonial stem cells, not with sperm. Besides, they didn't investigate Y-27632 localization or expression at different sperm areas.

A: Thank you for pointing out the problem. According to your comments and suggestions, we have replaced this reference.Line394:references22

We tried our best to revise manuscript. These changes will not influence the content and framework of the manuscript. We appreciate for editor’s and reviewers’ critical comments and thoughtful suggestions for our manuscript and hope that the revised manuscript will meet the standard of Animals.

Once again, thank you very much for your comments and suggestions.

Sincerely Yours,

Hongguo Cao

Reviewer 2 Report

Comments and Suggestions for Authors

I thank the team of authors for the experimental work done. Although the manuscript presents interesting results, I have a few criticisms:

1) Due to the fact that the manuscript discusses not only metabolomic regulation, but also the effect of Y-27632 on the quality parameters of goat sperm stored at low temperatures, I propose to reformulate the title of the manuscript (just as a variant: Positive in vitro effect of ROCK pathway inhibitor Y-27632 on qualitative characteristics of goat sperm stored at low temperatures)

2) In the “Collection and processing of semen” section, please provide more detail in what period of the year the samples were collected.

3) In the “Statistical analysis” section, please provide more details on how many biological and technical replicates were conducted. It is advisable to provide a general descriptive statistics.

4) In the section "Determination of energy metabolism in sperm" provide information on the model and technical characteristics of the flow cytometer used. For fig. 5 - 6, please provide control dot-plots as justification for the chosen gating strategy.

5) In the “Conclusion” section, please include a sentense about the need to verify the in vitro results obtained by in vivo experiments (i.e. using artificial insemination).

6) If possible, I ask to provide data on the combined effect of Y-27632 + the ROCK pathway activator AA in the revision (to confirm whether Y-27632 has a protective effect on sperm by affecting the ROCK pathway).

Once again I thank the authors for the work done. I hope the authors will take my criticism in a positive way.

Author Response

Dear Reviewers:

        We are truly grateful to you and the reviewers for the critical comments and thoughtful suggestions on our manuscript. They are really helpful and based on these comments and suggestions, we have revised the manuscript carefully. In the following pages are our point-by-point responses to the reviewers’ comments/suggestions. Please feel free to contact us if there is any question and we are very willing to improve our manuscript until all the reviewers are satisfied.

Best regards,

Hongguo Cao

Address: College of Animal Science and Technology, Anhui Agricultural University, Hefei 230036, P.R. China

E-mail: caohongguo1@ahau.edu.cn

Reviewer:2

C: Due to the fact that the manuscript discusses not only metabolomic regulation, but also the effect of Y-27632 on the quality parameters of goat sperm stored at low temperatures, I propose to reformulate the title of the manuscript (just as a variant: Positive in vitro effect of ROCK pathway inhibitor Y-27632 on qualitative characteristics of goat sperm stored at low temperatures)

A: Thank you for pointing out the problem. According to your comments and suggestions, we have revised the title of the paper. Line2-3

C: In the “Collection and processing of semen” section, please provide more detail in what period of the year the samples were collected.

A: Thank you for pointing out the problem. According to your comments and suggestions, we have added the detailed year of collecting sperm. Line104

C: In the “Statistical analysis” section, please provide more details on how many biological and technical replicates were conducted. It is advisable to provide a general descriptive statistics.

A: Thank you for pointing out the problem. According to your comments and suggestions, we have made modifications to this. Line227-230

C: In the section "Determination of energy metabolism in sperm" provide information on the model and technical characteristics of the flow cytometer used. For fig. 5 - 6, please provide control dot-plots as justification for the chosen gating strategy.

A: Thank you for pointing out the problem. According to your comments and suggestions, we have added the model of the flow cytometer. The reason for selecting the gating site map in Figures 5 and 6 is that the gating site map can clearly show the rise and fall of mitochondrial mode potential, directly reflecting the influence of Y-27632 and AA on mitochondrial membrane potential. Line175

C: In the “Conclusion” section, please include a sentense about the need to verify the in vitro results obtained by in vivo experiments (i.e. using artificial insemination).

A: Thank you for pointing out the problem. According to your comments and suggestions, we have added the results of artificial insemination. Line390-391

C:  If possible, I ask to provide data on the combined effect of Y-27632 + the ROCK pathway activator AA in the revision (to confirm whether Y-27632 has a protective effect on sperm by affecting the ROCK pathway

A: Thank you for pointing out the problem. This study investigated the effects of Y-27632 and AA on sperm, but did not investigate the comprehensive effect of Y-27632+AA.

We tried our best to revise manuscript. These changes will not influence the content and framework of the manuscript. We appreciate for editor’s and reviewers’ critical comments and thoughtful suggestions for our manuscript and hope that the revised manuscript will meet the standard of Animals.

Once again, thank you very much for your comments and suggestions.

Sincerely Yours,

Hongguo Cao
